# KVCache-Centric Memory for LLM Agents

## Abstract

LLM agents in complex, long-horizon workflows are constrained by the model's context window. Current plaintext-based memory systems suffer from unstable retrieval accuracy and disrupt prefix caching, harming both performance and efficiency. We propose MemArt, a novel memory paradigm that operates directly within the LLM-native format: the key-value (KV) cache. Instead of using plaintext, MemArt stores conversational turns as reusable KV cache blocks and retrieves relevant memories by computing attention scores in latent space. To enable accurate and efficient retrieval, we develop a multi-token aggregation retrieval strategy that uses compressed keys for efficient KV selection and a decoupled position encoding mechanism to ensure retrieved blocks are safely and coherently reused. On the LoCoMo benchmark, MemArt improves accuracy by over 11% (up to 39.4%) compared to state-of-the-art plaintext-based memory methods, nearly matching full-context performance. Critically, it achieves this while reducing prefill tokens by over two orders of magnitude (91-135×), representing a significant leap forward for building powerful and efficient long-context agents.

## 1 Introduction

> *"The true art of memory is the art of attention."* — Samuel Johnson, English writer

Large language model (LLM) agents are emerging as a new paradigm for applying foundation models in complex, real-world workflows, including scientific exploration (e.g., deep research (Xu & Peng, 2025)), coding assistants (Liu et al., 2024), and autonomous task planning systems (Wang et al., 2024a). Unlike single-turn prompting or short-lived chatbots, these agents are designed to operate over extended horizons, often spanning hours or days of execution and involving tens to hundreds of iterative LLM calls. During such long-running sessions, agents continuously accumulate rich context that quickly grows beyond the context window of even frontier models with million-token capacities (Google DeepMind, 2025). To address this scalability bottleneck, recent work has introduced external memory systems that store and selectively retrieve historical context (Chhikara et al., 2025; Rasmussen et al., 2025; Amazon Web Services, 2025). Such memory mechanisms are essential for sustaining reasoning efficiency, accuracy, and robustness in long-horizon agent workflows.

Most deployed memory systems, including Mem0 (Chhikara et al., 2025), Zep (Rasmussen et al., 2025), and AWS AgentCore memory (Amazon Web Services, 2025), adopt plaintext-based memory. They segment or summarize historical context into sentence-level memory entries, which are then indexed and retrieved using vector databases or graph structures. While straightforward, this approach exhibits two fundamental limitations. First, context summarization and retrieval based on vector similarity or graph traversal often fail to preserve the full semantic dependencies of long, multi-turn interactions. As a result, the retrieved memory may omit critical context or include irrelevant information, leading to degraded LLM inference performance compared to full-context inference. Second, the segmentation and summarization of historical context into discrete memory entries disrupts the natural sequential structure of prompt prefixes. Modern LLM engines

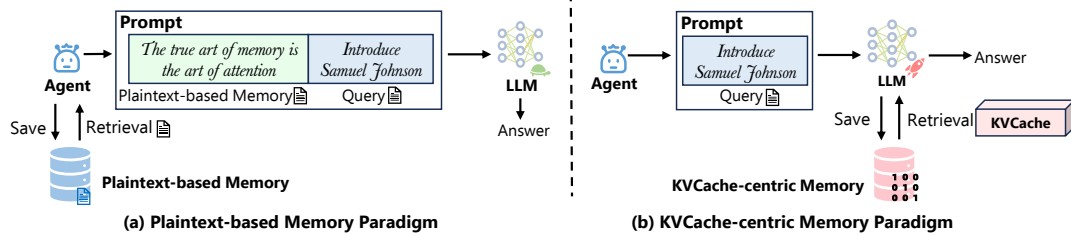

Figure 1: Paradigm comparison between plaintext-based memory and KVCache-centric memory. In the plaintext paradigm (a), the agent must explicitly retrieve and insert memory into the prompt, which often leads to inaccurate retrieval and breaks prefix caching. In the KVCache-centric paradigm (b), the LLM natively stores and reuses KV blocks, so the agent only issues the new query, enabling more accurate retrieval in latent space and efficient prefill reuse.

accelerate inference using prefix caching (Qin et al., 2025; Gao et al., 2024; Zheng et al., 2024)—reusing the key-value (KV) cache of shared prefixes across calls—but segmented and summarized memory introduces prefix discontinuities, undermining these efficiency gains.

We propose MemArt, a new memory paradigm that shifts from plaintext-based memory to *KVCache-centric memory* to enhance both performance and efficiency. As illustrated in Figure 1, instead of managing plaintext, MemArt stores historical context directly as reusable KV blocks and retrieves relevant memory by computing attention scores between the current prompt and the stored KV blocks in latent space. This approach offers three key advantages: (1) *High-Fidelity Retrieval:* Operating in latent space allows retrieval to align directly with the model's attention mechanism, offering superior semantic accuracy compared to methods relying on plaintext similarity. (2) *Maximal Inference Efficiency:* Retrieved KV blocks are directly reused during prefill, eliminating redundant token processing and significantly reducing computational overhead and latency. (3) *Seamless Integration:* The entire framework is model-agnostic and functions as a plug-and-play component, requiring no modifications to model weights or architecture.

Despite its promise, achieving KVCache-centric memory introduces two key challenges. First, *how can we perform high-fidelity retrieval without a full memory scan?* As the memory grows, exhaustively scanning every KV block to find the most relevant ones becomes computationally prohibitive. The challenge lies in designing a mechanism that can quickly identify the most salient memories from a large repository without sacrificing accuracy. Second, *how to ensure the safe reuse of retrieved KV blocks?* A standard KV cache that can reuse corresponds to a single, contiguous prefix. Retrieved blocks, however, are non-contiguous and carry their original positional information. Simply concatenating them creates a positionally incoherent sequence that disrupts the model's attention, ultimately harming output quality.

For efficient and accurate retrieval, MemArt first computes a compressed representative key for each KV block to enable a fast search that avoids a full memory scan. It then employs a multi-token aggregation retrieval strategy that synthesizes attention scores from all prompt tokens to ensure the final selection is highly relevant. For safe reuse, MemArt uses a decoupled position encoding mechanism. This component validates and adjusts the positional information of retrieved blocks, guaranteeing they can be integrated into the current context without creating positional conflicts.

We evaluate MemArt on the widely used LoCoMo benchmark (Maharana et al., 2024). Experimental results show that MemArt improves inference accuracy by 11.8–39.4% over state-of-the-art plaintext-based memory approaches, approaching the performance of full-context inference. Critically, it reduces 91–135× prefill tokens over plaintext-based memory approaches. These results highlight KVCache-centric memory as a promising foundation for accurate and efficient long-context LLM agents.

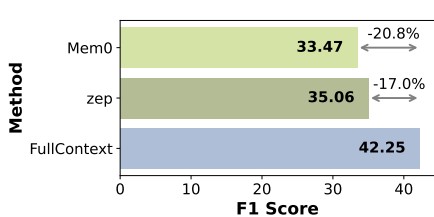 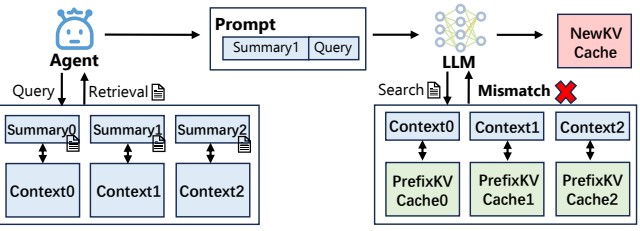

(a) Accuracy loss of plaintext-based memory retrieval on Qwen-2.5-7B-Instruct.

(b) Prefix cache reuse failure due to prefix context mismatch.

Figure 2: Limitations of plaintext-based memory: accuracy degradation and prefix cache reuse failure.

## 2 BACKGROUND AND RELATED WORK

**Prefix Caching in LLM Inference**   LLMs based on the Transformer architecture generate tokens autoregressively, with each token attending to all preceding tokens. To avoid redundant computation, the key (K) and value (V) tensors of previous tokens are stored as KVCache, enabling the prefill phase to cache K and V for the input prompt and the decode phase to generate new tokens by computing K and V only for the latest token. Building on this mechanism, prefix caching accelerates inference by sharing the KVCache of identical prefixes across requests, and has been widely adopted in recent systems to reduce computation and latency (Kwon et al., 2023; Zheng et al., 2024; Qin et al., 2025; Yao et al., 2025; Gao et al., 2024).

**Plaintext-Based Memory for LLM Agents**   Plaintext-based methods explicitly store and manipulate information in human-readable form.  Early systems such as MemoryBank (Zhong et al., 2024) and MemGPT (Packer et al., 2023) rely on predefined policies for storage, integration, and retrieval. Recent efforts shift toward structured representations, such as temporal knowledge graphs in Zep (Rasmussen et al., 2025), atomic notes in A-MEM (Xu et al., 2025), and hierarchical graph memories in Mem0 (Chhikara et al., 2025), which capture relational, temporal, and hierarchical dependencies but remain rule-based. Recent efforts also conceptualize memory as an operating system. MemoryOS (Kang et al., 2025) defines dynamic updates from short- to mid- to long-term memories, while MemOS (Li et al., 2025) defines unified representation, scheduling, and evolution across different memory types. However, MemOS does not provide detailed mechanisms for how these memory types coordinate or transform among each other.  In parallel, Memory-R1 (Yan et al., 2025) employs a reinforcement learning based manager to learn memory operations, though at the cost of significant training overhead.

**Limitations of Plaintext-Based Memory**   Plaintext-based memory suffers from two inherent limitations:

*1) Accuracy Degradation:* As shown in Figure 2(a), we evaluate the plaintext-based memory retrieval baseline Mem0 (Chhikara et al., 2025) and Zep (Rasmussen et al., 2025) on the LoCoMo benchmark with the Qwen-2.5-7B-Instruct model, which exhibit F1 gaps of 20.8% and 17.0%, respectively, compared to full-context inference. This highlights the difficulty of plaintext-based summarization and similarity retrieval in capturing long-range semantic dependencies. Critical information is frequently omitted, while irrelevant segments are introduced, both of which harm downstream reasoning.

*2) Prefix Caching Invalidation:* Prefix caching reuses the KV cache only when exact prompt prefixes match. However, plaintext memory systems typically segment, summarize, or alter historical context, breaking textual continuity. As illustrated in Figure 2(b), this textual mismatch prevents cache reuse and forces costly recomputation of the KV cache, undermining one of the most important efficiency gains in modern LLM inference.

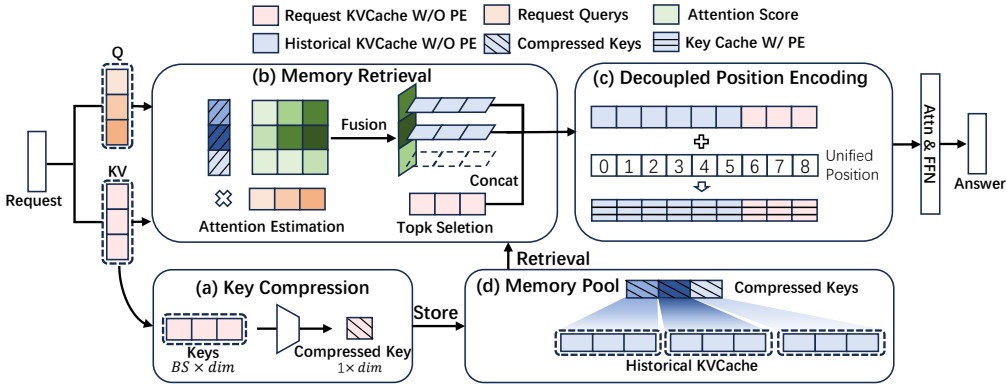

Figure 3: The architectural overview of MemArt.

## 3 MEMART: KVCACHE-CENTRIC AGENT MEMORY

To improve both performance and efficiency of agent inference, we propose MemArt, a new paradigm we term *KVCache-centric memory*. Instead of using plaintext, MemArt stores the KV cache as LLM-native memory. For each new request, MemArt identifies the most relevant historical KV blocks via latent-space attention and seamlessly reuses them for prefill. This design inherently yields three advantages: superior retrieval accuracy, dramatic gains in inference efficiency, and seamless plug-and-play integration. We now detail the MemArt framework and its core algorithmic components.

### 3.1 OVERALL FRAMEWORK

To enable efficient storage, retrieval, and reuse of KVCache-centric memory, MemArt adopts the framework shown in Figure 3, which comprises four key components:

a) **Key Compression:** Historical memory is stored as fixed-size KV blocks, each assigned a compressed key derived from its key set, providing a lightweight index that reduces retrieval overhead.

b) **Memory Retrieval:** For a new query request, attention scores are computed between the Q heads of all query tokens and compressed keys. The top-$k$ most relevant KV blocks are selected via multi-token aggregation retrieval and reused in the prefill phase.

c) **Decoupled Position Encoding:** Since historical memory may exceed the context window of LLMs, positional misalignment can weaken attention and degrade inference. To address this, KV blocks are stored without positional encodings and later re-embedded with new encodings after retrieval, ensuring alignment within the current context window and consistency for downstream attention.

d) **Memory Pool:** A centralized memory pool manages the collection of KV blocks, each indexed by its compressed key for efficient organization and access.

Building on these components, the inference workflow follows Algorithm 1: (i) retrieve query-relevant memory using compressed keys, (ii) concatenate the retrieved KV blocks with the KV cache of the query request, (iii) re-embed new positional encodings to align within the context window, and (iv) compute memory-augmented attention. Meanwhile, the memory pool is asynchronously updated with newly generated KV cache and its compressed keys. Formally, the process is given by:

$$O = \text{Attn}(\text{EmbPE}(Q, \text{Concat}(K_M, K)), \text{Concat}(V_M, V)) \tag{1}$$

where $K_M, V_M$ denotes the KV blocks retrieved from MemArt.

---

**Algorithm 1** Inference Workflow with MemArt

---

1: **Input:** Current Query $Q$, Current KVCache $(K_{curr}, V_{curr})$
2: **Memory:** Compressed Indices $CompK$, Memory Pool $(K_{mem}, V_{mem})$
   # Memory Augmented Generation
3: $(K_M, V_M) \leftarrow \text{Retrieve}(Q, CompK)$                          # Query-Aware Memory Retrieval
4: $K_{aug} \leftarrow \text{Concat}(K_M, K_{curr}), V_{aug} \leftarrow \text{Concat}(V_M, V_{curr})$          # Memory Integration
5: $Q', K'_{aug} \leftarrow \text{EmbPE}(Q, K_{aug})$                          # Align positional encodings
6: $O \leftarrow \text{Attention}(Q', K'_{aug}, V_{aug})$                          # Augmented Attention Computation
   # Memory Pool Update (Asynchronous)
7: $K_{mem} \leftarrow \text{Append}(K_{mem}, K_{curr}), V_{mem} \leftarrow \text{Append}(V_{mem}, V_{curr})$
8: $CompK \leftarrow \text{Append}(CompK, \text{Compress}(K_{curr}))$                          # Key Compression
9: **Return:** Output $O$

---

## 3.2 ALGORITHM DESIGN

This subsection details the design of the core components of MemArt, including the key compression algorithm, the memory retrieval strategy, and the decoupled positional encoding.

### 3.2.1 AABB-BASED KEY COMPRESSION

The generated KV cache is partitioned into memory blocks of size $BS$ and stored in a memory pool. To enable efficient retrieval, each key block $K$ is compressed into an axis-aligned bounding box (AABB) (Van den Bergen, 1997; Cai et al., 2014; Chen et al., 2024), defined by the maximum and minimum vectors that enclose all key vectors within the block, as shown in Equation 2. This compact representation preserves coarse-grained semantic information while avoiding exhaustive comparisons with individual key vectors, thereby enabling rapid and accurate retrieval.

$$
\begin{aligned}
s^{\min}(K) &= (\min_{i=1}^{BS} k_{i,1}, \min_{i=1}^{BS} k_{i,2}, \ldots, \min_{i=1}^{BS} k_{i,dim}) \in \mathbb{R}^{1 \times dim} \\
s^{\max}(K) &= (\max_{i=1}^{BS} k_{i,1}, \max_{i=1}^{BS} k_{i,2}, \ldots, \max_{i=1}^{BS} k_{i,dim}) \in \mathbb{R}^{1 \times dim}
\end{aligned}
\tag{2}
$$

AABB compression is well-suited for high-dimensional keys because it is lightweight ($2 \times dim$ values per block), preserves coordinate-wise extrema without distortions from projections like principal component analysis (PCA) (Hotelling, 1933), and provides a natural coarse-grained filter before fine-grained attention.

### 3.2.2 MULTI-TOKEN AGGREGATION BASED MEMORY RETRIEVAL

Our retrieval process operates at the block level, building on the principle that neighboring keys in a KV cache often share semantic importance (Jiang et al., 2024). To do this efficiently, we retrieve memory at the block level using the compressed keys introduced earlier. Following Arkvale (Chen et al., 2024), the relevance between a single query token $q \in \mathbb{R}^{1 \times dim}$ and block $K \in \mathbb{R}^{BS \times dim}$ is defined as the maximum dot product:

$$
I(q, K) = \sum_{i=1}^{dim} \max\left(q_i s_i^{max}(K),\, q_i s_i^{min}(K)\right)
\tag{3}
$$

This formulation provides an upper-bound estimate of the attention scores between $q$ and all keys within $K$, without exhaustively comparing each key. Consequently, $I(q, K_1) > I(q, K_2)$ indicates that block $K_1$ contains at least one key vector whose attention with $q$ exceeds that of every key in $K_2$.

Table 1: Formulations of normalization and aggregation strategies for multiple token relevance scoring.

| Normalization | Aggregation | Formula for $I(Q, K)$ | Notes |
|---|---|---|---|
| Softmax | Sum | $\displaystyle\sum_{q \in Q} \frac{\exp(I(q, K))}{\sum_{K' \in \mathcal{K}} \exp(I(q, K'))}$ | Sharp normalization; Balances all tokens. |
| Softmax | Max | $\displaystyle\max_{q \in Q} \frac{\exp(I(q, K))}{\sum_{K' \in \mathcal{K}} \exp(I(q, K'))}$ | Sharp normalization; Selects the strongest token. |
| Reciprocal Rank | Sum | $\displaystyle\sum_{q \in Q} \frac{1}{\text{rank}_q(K) + c}$ | Smooth normalization; Balances all tokens. |
| Reciprocal Rank | Max | $\displaystyle\max_{q \in Q} \frac{1}{\text{rank}_q(K) + c}$ | Smooth normalization; Selects the strongest token. |

Nevertheless, unlike the decoding scenario that Arkvale targets (Chen et al., 2024), which uses a single query token, agent memory retrieval occurs during prefill and must account for a multi-token prompt $Q$. This introduces a key challenge: relevance scores from different query tokens are not directly comparable, and different tokens may prioritize different memory blocks. A naive aggregation (e.g., averaging) would dilute these varied signals.

To create a unified relevance score for the entire prompt, we introduce a two-step aggregation procedure:

$$I(Q, \mathcal{K}) = \text{Agg}_{q \in Q} \left( \text{Norm}_{K \in \mathcal{K}} \big( I(q, K) \big) \right) \tag{4}$$

Here, $\mathcal{K}$ denotes the collection of all compressed keys. The process works as follows: *(1) Normalize per Token:* For each query token $q \in Q$, we first normalize its relevance scores $\{I(q, K) \mid K \in \mathcal{K}\}$ across all compressed keys. This crucial step makes the scores from different tokens comparable. *(2) Aggregate across Tokens:* Next, we aggregate these normalized scores across all query tokens to produce a single, final relevance score for each block.

We systematically consider several strategies for these two steps (Table 1). For Normalization, we can use *Softmax* to amplify the strongest signals or *Reciprocal-Rank* to create a smoother distribution that is less sensitive to outliers (Cormack et al., 2009). For Aggregation, we can employ *Sum* to weigh evidence from all tokens and *Max* to prioritize the single strongest token-block interaction.

Finally, based on the aggregated scores $I(Q, \mathcal{K})$, we select the top-$k$ memory blocks. For efficiency, the same set of $k$ blocks is selected for all attention heads. These blocks are then concatenated in their original chronological order to preserve temporal consistency for the final prefill computation.

### 3.2.3 DECOUPLED POSITIONAL ENCODING

A major challenge in reusing KV cache as long-term memory is the misalignment of positional embeddings across temporal spans. Cached key-value states with their original positional encodings can cause (i) incoherent attention when historical tokens' positions no longer match their locations in the reconstructed sequence, and (ii) positions exceeding the model's context window, leading to inference failure.

We address this by decoupling positional information from the stored KV cache. During storage, we omit the rotary positional encoding (RoPE) (Su et al., 2024) and preserve only content-dependent KV cache:

$$K_i^{\text{raw}} = W_k x_i, \quad V_i^{\text{raw}} = W_v x_i, \tag{5}$$

where $x_i$ is the hidden state of the $i$-th token, and $W_k, W_v$ are the key and value projection matrices. At inference, after retrieving top-$k$ memory blocks $\{K_{i_1}^{\text{raw}}, \dots, K_{i_K}^{\text{raw}}\}$ based on the current query $Q$, the memory

tokens are concatenated in historical order and re-encoded with a unified positional scheme:

$$\widetilde{Q}_j = R_{p(j)}Q_j^{\text{raw}}, \quad \widetilde{K}_j = R_{p(j)}K_j^{\text{raw}}, \quad \widetilde{V}_j = V_j^{\text{raw}}, \tag{6}$$

where $p(\cdot)$ is the absolute position in the concatenated sequence and the RoPE rotation matrix $R_p$. This ensures queries and historical memory share consistent positional information.

For example, if a query $Q$ contains three tokens and one memory block $\{K_{16}, K_{17}, \ldots, K_{23}\}$ is selected, after concatenation, the Ks in the memory block are reassigned positions $p(K) = [0, 1, \ldots, 7]$ and the query tokens have $p(Q) = [8, 9, 10]$. Applying RoPE re-encodes all tokens into a unified positional space, enabling coherent attention across memory and query.

## 4 EVALUATION

### 4.1 EXPERIMENTAL SETUP

**Datasets and Models** We adopt the LoCoMo benchmark (Maharana et al., 2024), a widely used suite for assessing long-term conversational memory in agent systems. It comprises 10 conversations, each containing an average of 589 dialogues and 13,960 words. To enable precise evaluation, each dialogue is paired with approximately 200 questions and their corresponding correct answers, allowing models to be tested on retrieving specific details from the full conversation history. We exclude the adversarial subset, as it does not provide ground-truth answers. We implement MemArt on top of HuggingFace Transformers (Wolf et al., 2020), and all experiments are conducted on LLaMA-3.1-8B-Instruct (Grattafiori et al., 2024), Qwen-2.5-7B-Instruct (Yang et al., 2025b) and Qwen-3-32B-A3B-Instruct (Yang et al., 2025a).

**Metrics** Following prior work (Chhikara et al., 2025; Li et al., 2025), we evaluate inference accuracy using two categories of metrics: lexical similarity and semantic correctness. Lexical similarity is measured with F1 Score (F1) and BLEU-1 (B1), which capture token-level overlap. Semantic correctness is measured with BERTScore-F1 (BERT) and cosine similarity (Sim) over sentence embeddings, reflecting meaning-level alignment. The average of these four metrics provides a more holistic measure of generation quality.

**Baselines** We compare MemArt against following representative baselines: (1) **Full-Context Inference:** The entire dialogue history is provided as input to the LLMs. (2) **Zep:** A retrieval-oriented agent that implements structured memory access strategies, enabling effective reasoning over temporally extended and multi-turn queries (Rasmussen et al., 2025). (3) **Mem0:** A modular memory architecture featuring explicit in-context memory operations, designed to facilitate scalable deployment while maintaining high retrieval fidelity (Chhikara et al., 2025). (4) **H2O:** A KV cache sparse approach that retains heavy-hitter keys to shrink the active KV set during current inference (Zhang et al., 2023).

### 4.2 MAIN RESULTS

We evaluate MemArt on the LoCoMo benchmark against Mem0 and Zep using LLaMA-3.1-8B-Instruct (L3), Qwen-2.5-7B-Instruct (Q2), and Qwen-3-32B-A3B-Instruct (Q3). To ensure fairness, we tune each method so that their KV lengths during decoding are comparable. MemArt uses a block size of 16 with top-$k$=128 (Softmax-Max) for L3 and Q3, and top-$k$=256 (RR-Max) for Q2. Mem0 is set to top-$k$=100 for L3/Q3 and 150 for Q2, while Zep uses top-$k$=20 for L3/Q3 and 25 for Q2. For the KV cache sparse method H2O, we configure it to use the same heavy KV cache budget as MemArt.

#### 4.2.1 PERFORMANCE COMPARISON

Table 2 shows the performance comparison of MemArt and the baselines on LoCoMo.

Table 2: Performance comparison between MemArt and baselines on the LoCoMo benchmark. Models are assessed on F1, B1, BERT, and Sim metrics (higher is better). Aver.S denotes the average across metrics. Best scores are in bold; second-best are underlined.

| Method | LLaMA-3.1-8B-Instruct | | | | | Qwen-2.5-7B-Instruct | | | | | Qwen-3-30B-A3B-Instruct | | | | |
|---|---|---|---|---|---|---|---|---|---|---|---|---|---|---|---|
| | F1 | B1 | BERT | Sim | Aver.S | F1 | B1 | BERT | Sim | Aver.S | F1 | B1 | BERT | Sim | Aver.S |
| FullContext | **48.12** | **40.34** | 64.24 | 74.10 | **56.70** | **42.45** | **35.79** | 61.52 | 70.97 | **52.68** | 51.06 | 43.65 | 66.29 | 76.21 | 59.30 |
| Zep | 30.60 | 24.67 | 45.76 | 64.98 | 41.50 | 35.06 | 28.14 | 56.60 | 66.72 | 46.63 | 42.23 | 36.17 | 61.19 | 72.15 | 53.18 |
| Mem0 | 26.86 | 21.60 | 53.18 | 60.93 | 40.64 | 33.47 | 28.36 | 57.68 | 66.55 | 46.51 | 34.96 | 28.85 | 58.16 | 68.10 | 47.51 |
| H2O | 38.48 | 30.90 | 59.38 | 68.42 | 49.29 | 29.44 | 22.32 | 54.03 | 62.51 | 42.07 | 40.77 | 32.43 | 59.86 | 69.16 | 50.55 |
| MemArt | 47.72 | 40.29 | **64.25** | **74.43** | 56.67 | 41.67 | 33.54 | **61.92** | **71.35** | 52.12 | **51.54** | **44.54** | **67.86** | **76.39** | **60.08** |

Table 3: Efficiency comparison between MemArt and baselines on the LoCoMo benchmark.

| Method | LLaMA-3.1-8B-Instruct | | Qwen-2.5-7B-Instruct | | Qwen-3-30B-A3B-Instruct | |
|---|---|---|---|---|---|---|
| | # Prefill Tokens | KV Length (Decode) | # Prefill Tokens | KV Length (Decode) | # Prefill Tokens | KV Length (Decode) |
| FullContext | 21,892 | 21,912 | 22,125 | 22,145 | 22,125 | 22,145 |
| Zep | 3,285 | 3,305 | 4,160 | 4,180 | 3,491 | 3,511 |
| Mem0 | 2,911 | 2,931 | 4,982 | 5,002 | 2,879 | 2,899 |
| H2O | 21,892 | 2,304 | 22,125 | 4,352 | 22,125 | 2,304 |
| MemArt | **32** | **2,100** | **37** | **4,153** | **37** | **2,105** |

MemArt reaches 99.9% of FullContext accuracy on L3, 98.9% on Q2, and 101.3% on Q3, showing that it nearly matches full-context generation while retrieving only a small fraction of memory. Relative to Zep and Mem0, MemArt improves accuracy by 36.5%/39.4% on L3, 11.8%/12.1% on Q2, and 12.9%/26.4% on Q3, respectively. Compare to KV Cache pruning method H2O, MemArt improves accuracy by 14.9% on L3, 23.9% on Q2, and 18.8% on Q3, respectively.

### 4.2.2 EFFICIENCY ANALYSIS

Table 3 reports the efficiency of MemArt against baselines on LoCoMo. Prefill efficiency is measured by the number of tokens computed per request, while decode efficiency is measured by the effective KV length.

For prefill, MemArt requires only 32 and 37 tokens on average for each request with LLaMA and Qwen, respectively, since it computes only the new query while reusing retrieved KV blocks. In contrast, plaintext-based methods (Zep, Mem0) must recompute all retrieved tokens, leading to 91–135× more prefill tokens. H2O, which computes the entire KV cache during prefill and defers pruning to decode, therefore incurs a similarly high prefill cost. Compared with FullContext inference and H2O, MemArt achieves an additional 598–684× reduction. For decoding, MemArt processes only the KV blocks selected for retrieval plus the current KV cache of query, reducing KV length by 0.6%–36.4% over Zep, Mem0 and H2O and by 81.2%–90.4% over FullContext.

End-to-end latency results in Figure 4 confirm these gains: despite KV selection and I/O overhead, MemArt delivers up to 2.30× and 2.38× speedup over Zep and Mem0, 13.70× over H2O and 9.9–15.8× speedup over FullContext.

### 4.2.3 ABLATION STUDY

Figure 5 evaluates different multi-token aggregation retrieval strategies in MemArt on LLaMA3.1-8B-Instruct and Qwen2.5-7B-Instruct. Each memory block is scored by normalizing per-token relevance $I(Q, \mathcal{K})$ (Table 1) and aggregating across query tokens. We compare Softmax vs. Reciprocal Rank (RR) normalization, Sum vs. Max aggregation, and varying block sizes. Results show that Max aggregation con-

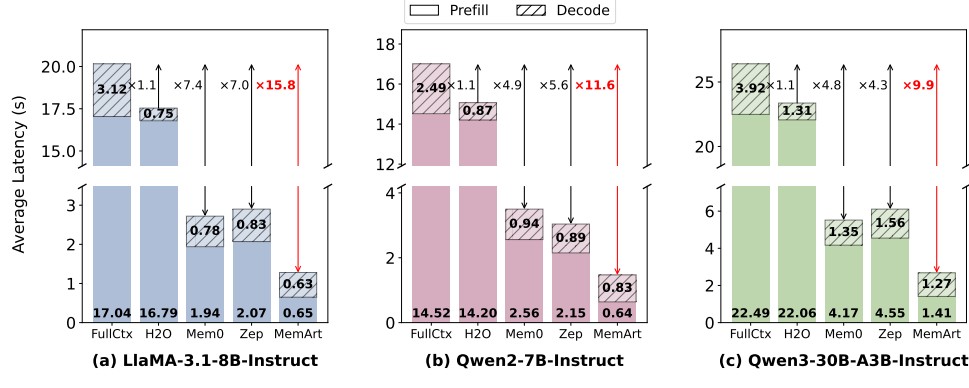

Figure 4: The average request execution latency of different methods.

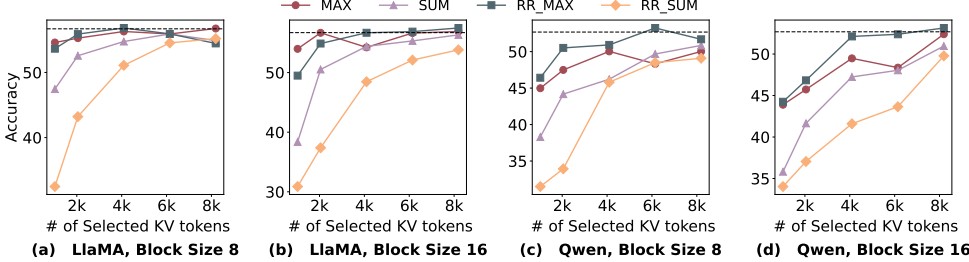

Figure 5: Ablation study of multi-token aggregation retrieval strategies.

sistently outperforms Sum, especially with fewer retrieved tokens, highlighting the benefit of emphasizing the strongest query-block interaction. Between normalization methods, Softmax and RR perform similarly on LLaMA, while RR-Max is superior on Qwen. Retrieval accuracy is largely insensitive to block size (8 vs. 16). In general, the choice of aggregation dominates the retrieval quality, while normalization and block size play secondary roles.

Table 4: Latency breakdown when storing KVCache in HBM vs. DRAM (Llama-3.1-8B-Instruct, 14k context).

| Method /Time(ms) | Attn | FFN | re-RoPE | Sel. TopK | Mem. Ops |
|---|---|---|---|---|---|
| Full Ctx | 10,072.7 | 5,397.2 | 0.0 | 0.0 | 0.0 |
| MemArt (HBM) | 17.9 | 23.5 | 5.5 | 117.5 | 29.9 |
| MemArt (DRAM) | 18.1 | 22.9 | 5.4 | 111.7 | 237.8 |

Table 4 details the latency breakdown of the prefilling phase when storing the 14k-token KV cache in HBM versus DRAM for the Llama-3.1-8B-Instruct model. The dramatic reduction in Attn and FFN latency is attributed to the significantly reduced computational scope: in contrast to the full context baseline ($14k \times 14k$), MemArt computes attention using a query length of 32 against a retrieved context of $32 + 2048$. This results in an over $2900\times$ reduction in FLOPs. Similarly, the FFN operates on a sequence length of only 32, compared to 14k in the baseline. The positional re-encoding adds a trivial 5.5 ms latency, while the cost of Top-K selection is acceptable due to our compressed key representation (AABB). Consequently, the primary latency bottleneck shifts to cross-tier memory transfers, specifically the migration of KV blocks from DRAM to HBM via PCIe. Note that these results reflect a naive, fully serialized implementation; this overhead can be substantially mitigated in future iterations through prefetching and overlapped compute.

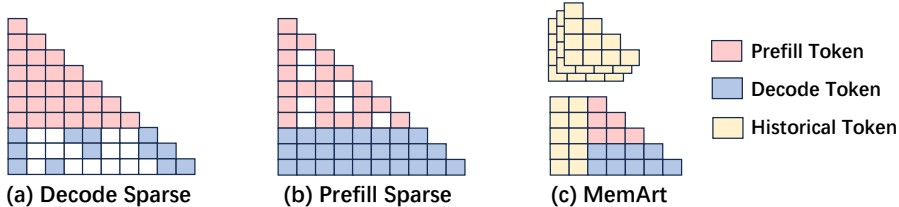

Figure 6: Comparison of attention mechanisms between MemArt and sparse attention.

# 5 DISCUSSIONS

**Differences from Sparse Attention Methods**    Although MemArt and sparse attention methods share the principle of utilizing the attention mechanism to identify salient information, they are fundamentally distinct. The primary differences lie in their temporal scope, core purpose, and the source of their attention scores. Specifically, sparse attention focuses on in-session acceleration by employing self-attention over the current context for sparsity, while MemArt targets memory-augmented generation by leveraging Query-Memory cross-attention over current and historical context for retrieval.

As shown in Figure 6, sparse attention generally falls into two categories. Decode sparse methods (e.g., H2O (Zhang et al., 2023), InfLLM (Xiao et al., 2024), Quest (Tang et al., 2024) and ArkVale (Chen et al., 2024)) maintain full attention during prefill but selectively compute important tokens at each decoding step. Prefill sparse methods (e.g., MInference (Jiang et al., 2024) and NSA (Yuan et al., 2025)) introduce sparsity by restricting each query token during the prefill stage to attend only to a subset of keys. The core goal of both categories is to accelerate the current inference speed or reduce the in-session KV Cache memory footprint, rather than to enhance long-horizon generation quality. Their sparsity is determined by self-attention over the current context window to infer which content is unimportant or redundant, thereby reducing computation.

In contrast, MemArt is an external, persistent long-term memory system whose primary goal is to significantly enhance the model's reasoning and generation quality. MemArt operates across a temporal divide: it performs retrieval once, prior to the prefill stage, injecting historical KV blocks from past sessions directly into the computation. Its attention scores are derived from cross-session cross-attention between the current request (Query) and the historical memory blocks (Keys), judging which historical information is most relevant and enhancing for the current inference. This essential spatio-temporal leap (historical data vs. current inference) and the blending of external retrieval with internal computation fundamentally distinguish MemArt from all sparse attention methods, providing the unique capability of long-horizon, non-contiguous memory recall.

# 6 CONCLUSION

This paper introduced MemArt, a paradigm that shifts agent memory from plaintext to the LLM's native KV cache. Our experiments on the LoCoMo benchmark show that this KVCache-centric approach is not only dramatically more efficient—reducing prefill tokens by over 90×—but also more accurate, improving accuracy by over 11%. Our work demonstrates that operating in the model's latent state is a more powerful and promising foundation for agent memory. Future research could explore learned KV cache compression and more sophisticated retrieval strategies. The code for MemArt will be made publicly available to encourage further exploration in this direction.

## REPRODUCIBILITY STATEMENT

We provide the necessary code, datasets, evaluation scripts, and the raw data used to derive our experimental conclusions in the supplementary material to ensure reproducibility.

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

# A APPENDIX

## A.1 STATEMENT ON THE USE OF LLMS

We used a large language model (ChatGPT) solely as a writing assistant to polish the language of this paper, such as improving grammar and clarity. The model was not involved in research ideation, methodological design, data analysis, or interpretation of results. All scientific content and conclusions were conceived and verified entirely by the authors.

## A.2 STRATEGY ABLATION ACROSS MULTIPLE METRICS

Figure 7 provides a granular view of the retrieval strategy ablation study, broken down by evaluation metric. The results reinforce the main findings presented in the body of the paper. We observe several consistent trends across all model and block-size configurations:

- Max aggregation is consistently superior to Sum, especially when retrieving a smaller number of tokens.
- On LLaMA, Softmax and Reciprocal-Rank (RR) normalization perform comparably, while the RR-Max combination yields the best results on Qwen.
- Retrieval accuracy shows low sensitivity to block size (8 vs. 16), suggesting the approach is robust.

Overall, the aggregation method is the dominant factor in retrieval quality, with normalization and block size playing secondary roles.

## A.3 THE IMPORTANCE OF DECOUPLED POSITIONAL ENCODING

The necessity of decoupling positional encodings for reliable long-term memory is illustrated in the case study shown in Figure 8. In this experiment, we load the system with a large historical context (approx. 1M tokens) and compare performance with and without our decoupling mechanism.

As shown, when reusing KV cache with the standard, coupled positional encoding, the model's positional awareness breaks down once the context length exceeds its native window size. This misalignment leads to a catastrophic failure, resulting in degenerative, repetitive text. In contrast, our decoupled positional encoding mechanism completely resolves this issue, enabling the model to correctly utilize information from the distant past and generate a coherent answer. This demonstrates that decoupling is not just an optimization but an essential component for enabling robust, long-term memory in KVCache-centric systems.

## A.4 INTERPRETABILITY CASE STUDIES

In this section, we provide the case study to illustrate how MemArt retrieves and uses historical KV blocks. This case consists of: (1) A KV-to-text-span mapping, showing the original textual segment corresponding

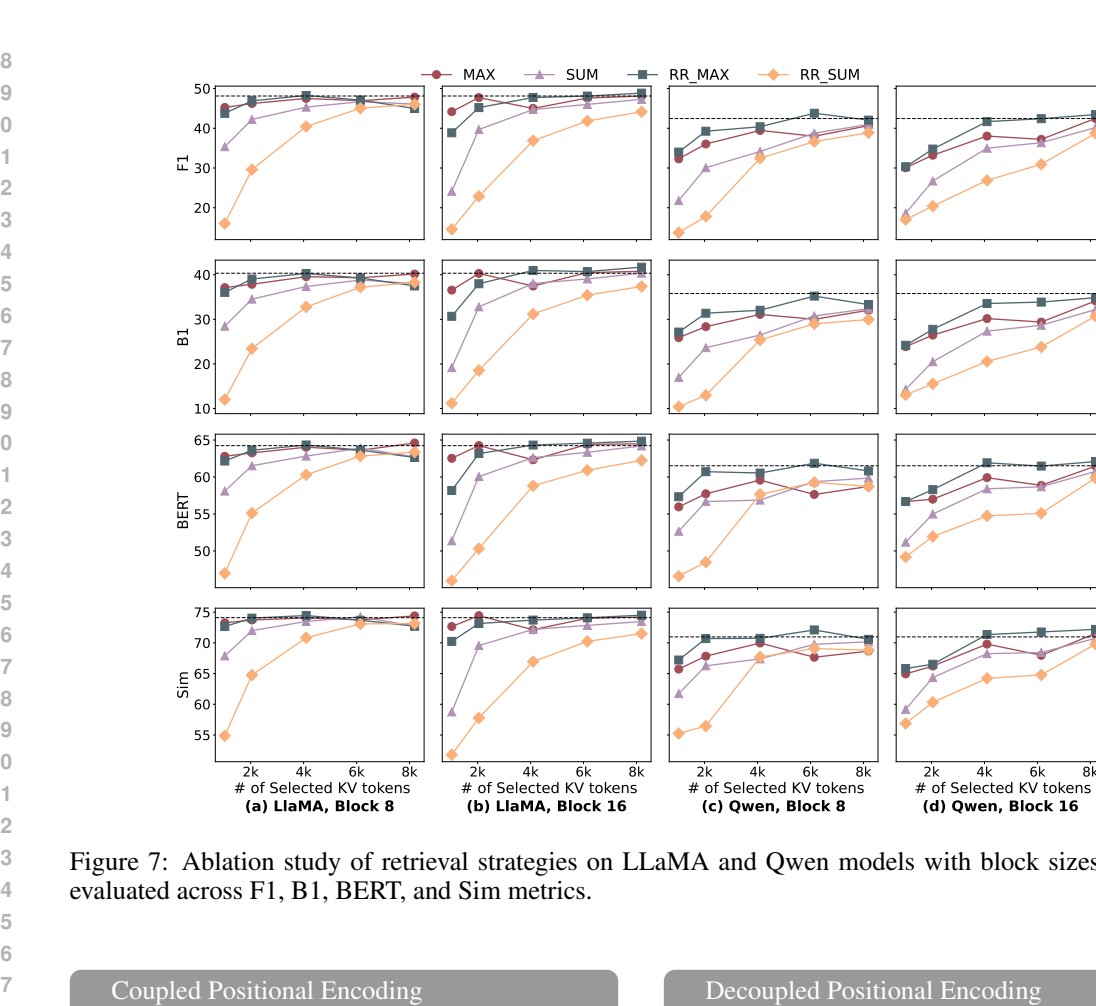

Figure 7: Ablation study of retrieval strategies on LLaMA and Qwen models with block sizes 8 and 16, evaluated across F1, B1, BERT, and Sim metrics.

| Coupled Positional Encoding | Decoupled Positional Encoding |
|---|---|
| **Context** Prior conversation of about 1M tokens. | **Context** Prior conversation of about 1M tokens. |
| **Question** What did Caroline research? Please answer the last question in few words and do not repeat the answer above: | **Question** What did Caroline research? Please7 answer the last question in few words and do not repeat the answer above: |
| **Answer** the the the the the the the the the the | **Answer** Adoption agencies. |

Figure 8: Case study on the importance of positional encoding decoupling for long-term memory.

to the retrieved KV block and (2) A layer-wise rank trajectory plot, showing how the relevance rank of this segment evolves across transformer layers.

**Retrieved Text Span**

**Query and Gold Answer**
Q: What did Caroline research?
A: Adoption agencies.

**Crucial Context Excerpt**
Caroline: Researching adoption agencies
— it's been ...

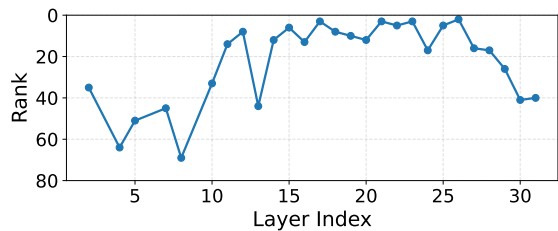

(a) KV-to-text-span mapping for the retrieved block.  (b) Layer-wise rank trajectory of the retrieved block.

Figure 9: Interpretability case study for the query *"What did Caroline research?"*. (a) textual span corresponding to a retrieved KV block. (b) layer-wise rank trajectory of the block.

Figure 9 presents a qualitative analysis designed to improve interpretability of MemArt's retrieval behavior. For the example query "What did Caroline research?", the retrieved block contains the phrase "Caroline: Researching adoption agencies — ..." shown in the left panel. The right panel plots the layer-wise rank trajectory of this KV block across the Llama's 32 transformer layers. The block begins with moderate relevance (e.g., ranks 35, 64, 51 of 920) but rapidly rises into the top ranks after Layer 10 and remains consistently within the top-5 most relevant blocks in deeper layers.

Together, the text-span visualization and rank trajectory show not only which block MemArt retrieves, but also how strongly and consistently the model relies on it across layers. This combined evidence demonstrates that MemArt retrieves semantically correct and persistent information rather than relying on incidental correlations.

### A.5 ROBUSTNESS OF MEMART UNDER ADVERSARIAL QUERIES

To evaluate the robustness of MemArt under adversarial, we conducted an additional experiment on the LoCoMo adversarial request subset using a strict factuality probing prompt: *"If the answer is not explicitly stated or directly inferable, respond only with: 'I don't know'."*

This setup tests whether the MemArt retrieval essential blocks required to avoid hallucination. On 100 adversarial queries, the full-context baseline produced 61 correct "I don't know" responses, while MemArt produced 57. The near-identical behavior indicates that our design preserves the relevant blocks even under adversarial queries and maintains generation correctness.

### A.6 MULTI-AGENT MEMORY COMPATIBILITY

MemArt can cleanly integrates into multi-agent environments, even when agents run heterogeneous models. The key idea is to preserve a model-agnostic communication channel across agents, while allowing each agent to maintain its own model-specific internal memory.

*Inter-agent sharing:* When heterogeneous agents need to communicate, they can exchange information through plaintext such as raw text, summaries, or structured memory, which remains model-agnostic.

*Intra-agent memory*: Once an agent consumes this information even once, it is encoded into its own KV cache and stored as MemArt blocks. Any future reuse of that information is purely an internal process. Subsequent retrieval relies solely on the agent's own representational space, enabling high-fidelity recall and compute efficiency without affecting cross-agent communication.

Thus, MemArt naturally coexists with multi-agent: plaintext memory remains the system-level lingua franca, while MemArt functions as a private, high-performance retrieval layer for each agent. This makes MemArt fully compatible with heterogeneous multi-agent systems: agents communicate externally in text, but internally obtain the benefits of KV-native long-term memory.

## A.7 ADDITIONAL EVALUATION ON PERSONAMEM

PersonaMem (Jiang et al., 2025) is a challenging long-horizon, multi-session dialogue benchmark specifically designed to assess an agent's ability to recall personalized, non-contiguous, and session-spanning information. Each instance features multi-turn conversations across several sessions, totaling an average context length of approximately 32K tokens. The task is evaluated using accuracy (correct answer rate) based on multiple-choice questions provided per instance.

Table 5: Accuracy comparison between MemArt and baselines on the PersonaMem benchmark. Best scores are in bold and second-best are underlined.

| Model | FullContext | Zep | Mem0 | H2O | MemArt |
|-------|-------------|-------|-------|-------|--------|
| L3 | **46.52** | 40.57 | 38.37 | 37.18 | 44.82 |
| Q2 | **54.50** | 47.20 | 47.37 | 48.73 | 52.97 |
| Q3 | **59.93** | 50.59 | 53.48 | 51.27 | 58.91 |

We evaluate MemArt against FullContext, Zep, Mem0, and H2O across Llama3.1-8B-Instruct(L3), Qwen2.5-7B-Instruct(Q2) and Qwen3-30B-A3B-Instruct(Q3) using the standard accuracy metric. Across all models, MemArt provides consistent and substantial gains over plaintext memory systems and KV-pruning baselines. Relative to Zep and Mem0, MemArt improves accuracy by 10.4%/16.8% on L3, 12.2%/11.8% on Q2, and 16.4%/10.1% on Q3, respectively. Compare to H2O, MemArt improves accuracy by 20.5% on L3, 8.7% on Q2, and 14.9% on Q3, respectively. Overall, these gains confirm that MemArt generalizes beyond LoCoMo and provides robust long-horizon recall for multi-session agent trace.

## A.8 SYSTEM-LEVEL CONSIDERATIONS: MEMORY I/O AND STORAGE

MemArt introduces a memory pool of historical KV blocks, which raises two system-level considerations: access latency and long-term storage growth. These concerns are inherent to all KV-centric serving architectures—not unique to our approach—as modern systems such as vLLM, SGLang, and Mooncake already persist and retrieve KV caches to enable prefix caching and reduce recomputation. MemArt follows the same principle but generalizes it to non-contiguous historical segments for long-horizon recall. Memory-pool I/O (HBM–DRAM transfers, prefetching, overlap with compute) is largely orthogonal to our algorithmic contributions; standard system optimizations such as asynchronous I/O, hierarchical caching, and double-buffering can be directly adopted. Similarly, MemArt is fully compatible with existing KV cache compression techniques (Devoto et al., 2024; Wang et al., 2024b), which can be layered on top to reduce storage footprint and further mitigate memory growth.