# OpenReview forum: "KVCache-Centric Memory for LLM Agents"
_ICLR.cc/2026/Conference — Submitted to ICLR 2026_

### Official Review · Reviewer_7mah · 2025-10-20

**Soundness:** 4
**Presentation:** 3
**Contribution:** 4
**Rating:** 4
**Confidence:** 4

**Summary:**

The paper proposes MemArt, a memory system that stores and retrieves past context directly in KV-cache space instead of plaintext. It introduces: (i) AABB-based key compression to index each KV block with min-max vectors; (ii) multi-token aggregation retrieval that scores blocks using normalized per-token relevance and then aggregates across tokens; and (iii) decoupled positional encoding that strips RoPE at storage time and re-embeds positions at reuse time to avoid positional mismatch.

**Strengths:**

MemArt’s KV-native retrieval aligns with the attention mechanism and removes prompt concatenation, which can avoid retrieval drift and preserve prefix-caching efficiency. The AABB compression is simple and allows a fast coarse filter before fine attention. The multi-token aggregation is well-motivated and the ablations (Softmax vs reciprocal-rank; Sum vs Max; block size) help isolate what matters. The decoupled positional encoding is clearly described and addresses long-context reuse failure modes.

**Weaknesses:**

1. Model coverage is limited for a 2025–2026 submission. Results are only on LLaMA-3.1-8B-Instruct and Qwen-2.5-7B-Instruct, with no newer families and no size sweep to show scaling trends.

2. Baseline breadth is narrow. The method is compared to plaintext memory systems (Mem0, Zep), but there is no head-to-head with cache-centric and dynamic sparse attention systems that also select KV blocks (for example Arkvale, InfLLM, Quest, NSA), even though they are discussed.

3. Scope of datasets is narrow. All main results are on LoCoMo; there is no test on other long-horizon agent traces

**Questions:**

Because MemArt stores and retrieves latent KV-cache tensors instead of text, the retrieved memory is not human-interpretable. This makes it difficult to verify what information is actually being recalled or whether retrieval errors occur. Can you provide any mechanism to improve interpretability — for example, storing lightweight metadata (token spans, summaries, or embeddings) alongside each KV block, or decoding retrieved KV tensors back into approximate text via the model’s unembedding layer? Additionally, can you report any qualitative analysis showing that the retrieved memories correspond to semantically relevant parts of the dialogue? Without such transparency, it is hard to assess whether MemArt retrieves correct information or merely benefits from implicit correlations.

---

> ### Author Response · Authors · 2025-11-21
> **Response to Reviewer 7mah**
>
> We appreciate your very careful reading and your positive assessment of the soundness and contribution. We summarize our responses and new results below.
>
>
> ##  **Weakness 1: Model Coverage and Scaling Trends.**
>
> We agree that the original experiments were limited in scale. Following your suggestion, we now include experiments on **Qwen3-30B-A3B-Instruct-2507** to study scaling behavior.
>
> The new results show: (1) MemArt achieves slightly *higher* average accuracy than FullContext (101.3\% of FullContext), likely because retrieval focuses attention on the most relevant memory. (2) MemArt consistently outperforms Zep, Mem0, and H2O while preserving its large prefill-token reductions.
>
> || FullContext | Zep   | Mem0  | H2O   | MemArt |
> |-| - | - | - | - | - |
> |Acc.| 59.30       | 53.18 | 47.51 | 50.55 | 60.08  |
> | \# Prefill Tokens  | 22,125      | 3,491 | 2,879 | 22,125 | 37     |
> | KV Length (decode) | 22,145      | 3,511 | 2,899 | 2,304  | 2,105  |
>
> These results, together with the original 7B/8B models, demonstrate that the MemArt paradigm scales robustly to larger, newer model families.
>
>
>
> ## **Weakness 2:  Missing Comparisons with Sparse Attention Systems.**
>
> Thank you for this crucial suggestion. Before adding empirical comparisons, we first clarify the conceptual distinctions in the revised  *Discussion* section, and then describe the new experiments we have added.
>
> The primary differences of MemArt and sparse attention system lie in their temporal scope, core purpose, and the source of their attention scores. Sparse attention focuses on **in-session acceleration** by employing **self-attention** over the current context to prune computation and memory overhead. In contrast, MemArt serves as a persistent long-term memory system. It operates before the prefill stage, leveraging Query-Memory **cross-attention** over current and historical context to retrieve non-contiguous KV blocks from **past sessions**. This mechanism is explicitly designed to strengthen the model’s reasoning and enhance generation quality by augmenting the context with historical information, a capability that sparse attention methods are not built to provide.
>
> To empirically address your concern, we incorporated H2O as a strong and widely-used KV sparse baseline.  The new results show that  MemArt is significantly more accurate (up to 23.9\% improvement) on this long-horizon agent task.
>
> | Method | L3    | Q2    | Q3    |
> | - | - | - | - |
> | H2O    | 49.29 | 42.07 | 50.55 |
> | MemArt | 56.67 | 52.12 | 60.08 |
>
> ## **Weakness 3: Dataset Scope.**
>
> We agree that relying solely on LoCoMo is limiting. Following your suggestion, we added evaluations on an additional long-horizon, multi-turn dataset, **PersonaMem[1]**.
>
> The new results are revised in Appendix A.7:
>
> | Models | FullContext | Zep   | Mem0  | H2O   | MemArt |
> | - | - | - | - | - | - |
> | L3     | 46.52       | 40.57 | 38.37 | 37.18 | 44.82  |
> | Q2     | 54.50       | 47.20 | 47.37 | 48.73 | 52.97  |
> | Q3     |59.93      | 50.59 | 53.48 | 51.27 | 58.91  |
>
> Across models, MemArt consistently tracks FullContext and improves over  Zep, Mem0, and H2O. This additional dataset supports that MemArt generalizes beyond a single benchmark and is applicable to broader long-horizon personalization scenarios.
>
> [1] Jiang et, al. "Know Me, Respond to Me: Benchmarking LLMs for Dynamic User Profiling and Personalized Responses at Scale", arXiv:2504.14225
>
> ##  **Question: Interpretability of Retrieved KV Blocks.**
>
> We fully share your concern that KV tensors are not directly human-interpretable. To address this, we added two interpretability mechanisms and a qualitative case study.
>
> **(1) KV-to-text-span mapping.**  During memory construction, we store lightweight metadata that maps each KV block to its corresponding text span(s). In Appendix A.4, we present examples where the retrieved blocks align with semantically relevant segments of the dialogue.
>
> **(2) Layer-wise ranking trajectories.** We further track how the relevance of key sentences evolves across layers by recording their rank among all memory blocks. For example, for the query ``What did Caroline research?`` (gold answer: ``Adoption agencies.``), MemArt repeatedly retrieves the segment ``Caroline: Researching adoption agencies ...``: its rank improves from 35 (Layer 1) to 8 (Layer 11) to 3 (Layer 16) and remains among the top candidates in deeper layers. This cross-layer stability indicates that MemArt is not relying on incidental correlations, but consistently focuses on the correct memory.
>
> These visualizations make it clear that MemArt's retrieved KV blocks correspond to meaningful, task-relevant historical information.
>
>
>
> We thank you again for your constructive feedback. The extended model coverage , additional baselines , broader dataset evaluation , and new interpretability analyses substantially strengthen the paper and directly address your concerns. We hope these revisions will positively inform your final assessment.

---

> > ### Comment · Reviewer_7mah · 2025-11-25
> >
> > Thanks for the clarification. For the model scaling, it seems that your experiments only went up to 30B. Do you have any results or expectations for larger models such as 70B? Also, since your method is built around open-source models, how do you think the impact changes if you cannot evaluate it on the strongest closed-source LLMs?

---

> > > ### Author Response · Authors · 2025-11-28
> > > **Response to Reviewer 7mah**
> > >
> > > ### **Result for 70B Models**
> > >
> > > Following your suggestion, we have added experiments on **Llama3.1-70B-Instruct** using the LoCoMo dataset to further examine scaling behavior. The results show that MemArt (*RR\_MAX, block\_size=16, topk=128*) consistently outperforms both Mem0 and Zep across all metrics, and even achieves a *slightly higher* average score than the FullContext baseline (101.1\%). These findings demonstrate that the MemArt paradigm scales robustly to larger models and maintains strong memory-retrieval quality.
> > >
> > > | Method       | F1     | B1     | BERT-F1 | Cosine Sim | Aver. Score | Ratio |
> > > | ------------ | ------ | ------ | ------- | ---------- | ----------- | ----- |
> > > | FullContext | 0.5070 | 0.4289 | **0.6602** | 0.7471     | 0.5858      | 100\%  |
> > > | Mem0 | 0.3488 | 0.2784 | 0.5693 | 0.6742  | 0.4676   | 79.8\% |
> > > | Zep | 0.3976 | 0.3136 | 0.5934 | 0.7086  | 0.5033   | 85.9\% |
> > > | **MemArt** | **0.5115** | **0.4417** | 0.6592 | **0.7566** | **0.5922** | **101.1\%** |
> > >
> > > ### **Impact on closed-source LLM**
> > >
> > > Stronger models generally possess more precise attention distributions, which directly enhance our attention-guided memory retrieval mechanism. Consequently, we anticipate that our method will not only maintain robustness but potentially improve as model size increases. This expectation is empirically supported by our scaling experiments (7/8B → 30B → 70B), where MemArt demonstrates consistent stability. Furthermore, as parameter counts rise, the efficiency advantage of MemArt becomes increasingly pronounced. Given these factors, we are confident that our approach will transfer effectively to closed-source LLMs.
> > >
> > > We hope these additional results and clarifications help address your concerns, and we would greatly appreciate your reconsideration of our assessment.

---

### Official Review · Reviewer_A26o · 2025-11-01

**Soundness:** 3
**Presentation:** 2
**Contribution:** 2
**Rating:** 6
**Confidence:** 3

**Summary:**

MemArt reframes agent memory as KVCache-centric rather than plaintext. The paper shows how MemArt stores past turns as reusable KV blocks and retrieves them by computing attention in latent space. This avoids retrieval drift and preserving prefix-caching benefits.
The system comprises (a) AABB-based key compression for each fixed-size block, (b) multi-token aggregation retrieval that scores blocks against all query tokens, (c) decoupled positional encoding that re-embeds retrieved KV without stale RoPE offsets, and (d) a managed memory pool. Compression represents each block with coordinate-wise minima and maxima, enabling fast coarse filtering. Notably, the relevance for a single token is upper-bounded via the dot-product with those bounds.  For multi-token prompts, scores are first normalized per token across all blocks and then aggregated to select top-k blocks in chronological order. Retrieved blocks are concatenated with the current KV and re-encoded with unified positions, ensuring coherent attention within the current window without exceeding native limits.

**Strengths:**

MemArt's design is quite interestng. It reframes memory to be KVCache-centric with latent-space retrieval, decoupled positional encoding, and lightweight AABB key compression with multi-token aggregation. This yields a model-agnostic, plug-and-play system.

**Weaknesses:**

System-wise, memory-pool I/O can add non-trivial latency, and safe reuse critically depends on decoupled positional re-embedding. The issue is that, without it, long-context behavior can be non-performant.

**Questions:**

1. I am curious, what is the precision and recall trade-off of the AABB block filter on adversarial or highly paraphrased queries?

2. What is the end-to-end latency and memory-traffic breakdown (prefill, retrieval, re-embed), and how would specialized KV hardware change the bottlenecks?

3. How does MemArt compare head-to-head with KV pruning strategies like Keyformer and MorphKV under the same memory budget and latency constraints?

---

> ### Author Response · Authors · 2025-11-21
> **Response to Reviewer A26o**
>
> Thank you for your thoughtful review and for highlighting both the novelty of the KVCache-centric design. Below we address your concerns regarding system-level considerations, robustness of the AABB filter, latency breakdown, and comparisons with KV-pruning strategies.
>
> ## **Weakness: System Level Consideration.**
>
> We thank you for raising this important system-level concern. We would like to clarify two points:
>
> (1) **Memory-pool I/O.** MemArt focuses on the *algorithmic feasibility* of KV-native memory: AABB-based compression, multi-token aggregation, and decoupled positional encoding. The I/O path (HBM-DRAM, asynchronous prefetch, overlap with compute) is largely orthogonal and shares the same challenges as existing KV-centric serving systems (prefix caching).  We make this separation explicit in the revision *Appendix A.8*.
>
> (2) **Cost of RoPE re-embedding.**  We added a latency breakdown (see Table 4 in the experimental section). It shows that positional re-embedding indeed has small overhead ($\le$ 6 ms) relative to attention / FFN and KV transfer, confirming that decoupling RoPE is both necessary for correctness and inexpensive in practice.
>
> ## **Question 1: Robustness of AABB filter.**
>
> To evaluate this, we conducted an additional experiment using the adversarial request subset from LoCoMo with a strict probing prompt: *"If the answer is not explicitly stated or directly inferable with certainty from the given information, respond with only: 'I don't know'."* This assesses MemArt's ability to maintain correct negative assertions, which is a key measure of robustness.
>
> Across 100 adversarial queries, the full-context baseline correctly answered *“I don’t know”* 61 times, while MemArt, utilizing the AABB filter, maintained this decision 57 times. This near-identical behavior indicates that the AABB coarse filter exhibits excellent robustness against adversarial queries.  This analysis is included in the Appendix A.5.
>
>
>
> ## **Question 2: End-to-End Latency breakdown.**
>
> We added a detailed end-to-end breakdown (prefill, retrieval, re-embedding, KV transfer). A representative example is:
>
> | Attn+FFN | re-embed | topk retrieval | Memory Ops in HBM | Memory Ops in DRAM |
> | - | - | - | - | - |
> | 41 ms    | 6 ms     | 111 ms         | 30 ms             | 238 ms             |
>
> The results show that: (1) MemArt removes nearly all full-context compute, since the attention computational scope reduces from 14k$\times$14k to 32$\times$(32+2048). (2) The critical decoupled positional re-embedding adds a negligible 6 ms overhead. (3) The Top-K retrieval, which includes the AABB coarse filter and multi-token aggregation, requires 111 ms. (4) When KV blocks are in HBM, memory traffic (concat) costs only 29.9 ms; when stored in DRAM, the transfer cost rises to 237.8 ms, becoming the primary bottleneck. However, the I/O results reflect a naive, fully serialized implementation; this overhead can be substantially mitigated in future iterations through prefetching and overlapped compute.
>
> ## **Question 3: Comparison with KV-Pruning Strategies.**
>
> We appreciate your suggestion. Before discussing empirical results, we clarified in the revised *discussion* section that MemArt and KV-pruning systems operate under different goals and mechanisms:
>
> The primary differences lie in their temporal scope, core purpose, and the source of their attention scores. KV-pruning systems (e.g. H2O, Keyformer and MorphKV ) focuses on **in-session acceleration** by employing **self-attention** over the current context to prune computation and memory overhead. In sharp contrast, MemArt serves as a persistent long-term memory system. It operates before the prefill stage, leveraging Query-Memory **cross-attention** over current and historical context to retrieve non-contiguous KV blocks from **past sessions**. This mechanism is explicitly designed to strengthen the model’s reasoning and enhance generation quality by augmenting the context with historical information, a capability that sparse attention methods are not built to provide.
>
> We added H2O as a representative KV pruning baseline in our experiments. H2O captures the same design philosophy as Keyformer/MorphKV by pruning the active KV entries. The new results show that  MemArt's retrieval-focused design is significantly more accurate (up to 23.9\% improvement) on this long-horizon agent task with same memory budget.
>
> | Method | L3    | Q2    | Q3    |
> | - | - | - | - |
> | H2O    | 49.29 | 42.07 | 50.55 |
> | MemArt | 56.67 | 52.12 | 60.08 |
>
> Thank you again for your detailed questions. The additional robustness study, latency breakdown, and new baseline comparison directly address your concerns and, we believe, strengthen the system-level story. We hope these revisions will positively inform your final assessment.

---

> > ### Comment · Reviewer_A26o · 2025-11-27
> >
> > Thank you. This answers most of my questions.

---

### Official Review · Reviewer_ja5f · 2025-11-02

**Soundness:** 2
**Presentation:** 3
**Contribution:** 2
**Rating:** 4
**Confidence:** 4

**Summary:**

This paper proposes MemArt, a new KV-cache centric memory paradigm for LLM agents that replaces plaintext with direct reuse of latent KV cache blocks. Instead of re-feeding retrieved text into prompts, MemArt stores and retrieves prior computation directly in latent space which dramatically improves both accuracy and efficiency. Specifically, they propose to compress keys via a bounding box, then they compute the attention over KV blocks through normalization and aggregation over the query tokens. Finally, they append these KV blocks after injecting the positional index to start the decoding.

**Strengths:**

* The proposed KV-cache centric memory paradigm can directly reuse the calculated KV during prefill, which reduces computational overhead.
* The proposed multi-token aggregation does alleviate retrieval overhead by reducing the number of index.
* Their proposed decoupled positional encoding practically solves the issue.

**Weaknesses:**

* While the proposed method achieves higher accuracy and lower latency, it inevitably involves an ever growing memory size that might cause storage issue. This is due two design choices: 1) the KV cache is represented in float numbers and it scales much faster than plaintext; 2) the memory is linearly growing with no upper bound on the size.
* Another drawback of using KV cache paradigm is the generalization across models. The importance of memory sharing amplifies in multi-agent systems, where one model needs to understand the other model’s memory. It limits the scope of the paper.
* There seems to lack some experimental comparison with KV cache compression literature. I have listed several below for reference.
1. _H2O: Heavy-Hitter Oracle for Efficient Generative Inference of Large Language Models_
2. _SnapKV: LLM Knows What You are Looking for Before Generation_
3. _A Simple and Effective L2 Norm-Based Strategy for KV Cache Compression_
* Finally, the model size experimented are limited and primarily lies around 7/8B. I believe the work benefit from validating on larger scale models such as 32B or MoE models.

**Questions:**

* How would you discriminate your work from KV cache compression literature?

---

> ### Author Response · Authors · 2025-11-21
> **Response to Reviewer ja5f**
>
> Thank you for your constructive and insightful review. Here is a point-by-point response to your concerns:
>
> ## **Weakness 1: Memory Growth.**
>
> We agree that storage growth is a crucial concern for long-term deployment. Importantly, MemArt does not introduce a fundamentally new storage assumption: it builds on the same premise as prefix caching widely used in existing LLM serving systems, such as vLLM and SGLang, which persist historical KV cache and reuse it across requests to reduce prefill overhead.
>
> MemArt not only covers but fundamentally extends the capability of prefix caching. Standard Prefix Caching can only reuse continuous and identical prefixes. MemArt, through our novel retrieval mechanism, enables the reuse of arbitrary, non-contiguous KV blocks from history. Therefore, MemArt does not introduce a new storage problem; it leverages and significantly enhances the industry-standard practice of storing the KV cache to boost agent memory accuracy.
>
> Moreover, MemArt can also be combined with KV cache compression policies to reduce storage overhead. We add this discussion in the *Appendix A.8*.
>
>
> ## **Weakness 2: Generalization Across Models and Multi-Agent Sharing.**
>
> We agree that KV cache is model-specific and that cross-model memory sharing is important in multi-agent settings. Our goal with MemArt is not to replace all forms of memory, but to serve as a **high-performance intra-agent memory** layer that is *complementary* to plaintext-based inter-agent sharing.
>
> *Inter-agent sharing*: When heterogeneous agents  need to communicate, they can exchange information through plaintext (raw text, summaries, or structured memory), which remains model-agnostic.
>
> *Intra-agent memory*: Once an agent consumes this information even once, it is encoded into its own KV cache and stored as MemArt blocks. From this point on, all subsequent retrieval for that agent uses MemArt, which maximizes its *own* accuracy and efficiency.
>
> Thus, MemArt naturally coexists with multi-agent plaintext memory: plaintext remains the lingua franca across agents, while MemArt acts as an internal accelerator that gives each agent a KV-native, high-fidelity memory. We clarify these complementary roles in the revised *Appendix A.6* section.
>
>
>
>  ##  **Weakness 3 \& Question 1: Comparison with KV Cache Compression.**
>
> Thank you for prompting us to clarify this distinction. Our work is fundamentally different from, though complementary to, KV cache sparse (H2O and SnapKV) and compression(L2).  We clarify this in the revised *Discussion* section.
>
> The primary differences lie in their temporal scope, core purpose, and the source of their attention. Sparse attention(e.g. H2O, SnapKV) focuses on **in-session acceleration** by employing **self-attention** over the current context to prune computation and memory overhead. In contrast, MemArt serves as a persistent long-term memory system. It operates before the prefill stage, leveraging Query-Memory **cross-attention** over current and historical context to retrieve non-contiguous KV blocks from **past sessions**. This mechanism is explicitly designed to strengthen the model’s reasoning and enhance generation quality by augmenting the context with historical information, a capability that sparse attention methods are not built to provide.
>
> Moreover, MemArt is orthogonal to L2 norm–based KV-cache compression methods, which can be jointly used with MemArt to mitigate storage overhead.
>
> *New Results:* Following your suggestion, we run a new experiment comparing MemArt to a strong KV cache eviction baseline (H2O) on LoCoMo. The results show that MemArt consistently outperforms H2O on the agent task:
>
> | Method | L3    | Q2    | Q3    |
> | - | - | - | - |
> | H2O    | 49.29 | 42.07 | 50.55 |
> | MemArt | 56.67 | 52.12 | 60.08 |
>
>
>
> ##  **Weakness 4: Limited Model Scale**
>
> We appreciate this suggestion and have extended our evaluation beyond 7B/8B models. We now include experiments on **Qwen3-30B-A3B-Instruct-2507** for our main results (Tables 2 and 3).
>
> *New Results:* On LoCoMo, MemArt **Matches or slightly exceeds** FullContext accuracy (101.3\% of FullContext average score), and continues to improve over plaintext baselines by a clear margin, while preserving its large prefill-token reduction.
>
> || FullContext | Zep   | Mem0  | H2O   | MemArt |
> |-| - | - | - | - | - |
> |Acc.| 59.30       | 53.18 | 47.51 | 50.55 | 60.08  |
> | \# Prefill Tokens  | 22,125      | 3,491 | 2,879 | 22,125 | 37     |
> | KV Length (decode) | 22,145      | 3,511 | 2,899 | 2,304  | 2,105  |
>
>
> These results show that MemArt's benefits *scale* with larger models. We highlight this scaling trend in the revised experimental section 4.2.1.
>
>
>
> Thank you again for your helpful comments. We believe the new experiments (H2O comparison, 30B model evaluation) and added clarifications substantially strengthen the paper and directly address your concerns. We hope the new results and analyses will positively inform your final assessment.

---

> ### Author Response · Authors · 2025-11-28
>
> Dear Reviewer ja5f,
>
> We hope this message finds you well. As the discussion period is nearing its end, we want to ensure that we have addressed all your concerns satisfactorily. If there are any additional points or feedback you would like us to consider, please let us know. Your insights are invaluable to us, and we are eager to address any remaining issues to further improve our work.
>
> Thank you very much for your time and effort in reviewing our paper.
>
> Best regards,
>
> The Authors of Paper 11052

---

### Author Response · Authors · 2025-12-01
**Summary Comment for Paper 11052 (1/2)**

We sincerely thank all reviewers for their constructive feedback. During the rebuttal period, we conducted substantial revisions and extensive new experiments to fully address the primary concerns raised by Reviewers. Below, we summarize the major revisions made in response to each reviewer’s concerns.

---

### **1.Reviewer ja5f (Initial Rating: 4)**

#### **Key Concerns**

* Storage growth of KV-cache–based memory
* Generalization across models, particularly in multi-agent settings
* Conceptual distinction between MemArt and KV-sparse / KV-compression methods
* Lack of larger model evaluation

#### **Our Responses & Revisions**

* **KV-Cache Storage Issue**

  We clarified that MemArt inherits the same storage assumption as **prefix caching** widely deployed in vLLM/SGLang. While MemArt significantly enhances this industry-standard practice of storing the KV cache to boost agent memory accuracy, and can integrate with existing KV compression  to reduce storage overhead. We add this discussion in the *Appendix A.8*.

* **Multi-agent Memory Compatibility**

  We clarified that our goal with MemArt is not to replace all forms of memory, but to serve as a **high-performance intra-agent memory** layer that is *complementary* to plaintext-based inter-agent sharing. We discuss these complementary roles in the revised *Appendix A.6* section.

* **Comparison with KV Cache Sparse/Compression**

  We clarified that MemArt is fundamentally distinct from sparse attention methods (which focus on **in-session** acceleration via self-attention over the current context) by operating as a persistent long-term memory system that uses **cross-session** Query-Memory cross-attention to enhance generation quality.  Moreover, we emphasized the orthogonality to KV-compression techniques. For empirical results, we conducted a new baseline of KV sparse method **H2O**.  MemArt improves accuracy by up to **23.9\%** under the same memory budget.

* **New Experiments for Model Scaling**

  We added new experiments on **Qwen3-30B-A3B-Instruct (MOE)** and **Llama3.1-70B-Instruct** to explore scaling trend. MemArt matches or exceeds FullContext (up to **101.3\%**) and consistently outperforms baselines.

#### **Current Status**

Although we did not receive any follow-up from Reviewer ja5f,  we believe all concerns regarding scaling, baselines, and conceptual clarification were fully addressed with the new experiments and detailed analysis.

---

### **2.Reviewer A26o (Initial Rating: 6)**

#### **Key Concerns**

* System overhead and latency breakdown
* Robustness of the AABB filter against adversarial queries
* Comparison with KV pruning/sparse method

#### **Our Responses & Revisions**

* **System Latency Decomposition**

  We provided a complete and quantified full latency decomposition of the MemArt workflow. The result shows that re-embedding overhead is negligible (**6ms**), and the retrieval (**111ms**) and I/O costs (**238ms**) are highly acceptable when weighed against the massive elimination of full-context compute time (**>10,000ms**). This analysis is detailed in the revised *§4.2.3* .

* **Robustness on Adversarial Queries**

  We tested MemArt on adversarial LoCoMo queries (revised in *Appendix A.5*). MemArt closely matches FullContext’s negative-answer accuracy, confirming that the AABB coarse filter exhibits excellent robustness against adversarial queries.

* **Comparison with KV Sparse Method**

  We clarified the fundamental difference between MemArt and KV pruning/sparse method and conducted  **H2O** as a representative KV sparse baseline.  MemArt improves accuracy by up to **23.9\%** under the same memory budget.

#### **Current Status**

Reviewer A26o responded that our clarifications have resolved most of their concerns and no more comment.

---

> ### Author Response · Authors · 2025-12-01
> **Summary Comment for Paper 11052 (2/2)**
>
> ### **3.Reviewer 7mah (Initial Rating: 4)**
>
> #### **Key Concerns**
>
> * Model coverage and scaling trends
> * Missing comparisons with KV sparse method
> * Limited dataset scope
> * Required interpretability of retrieved KV blocks
> * Impact on closed-source LLMs
>
> #### **Our Responses & Revisions**
>
> * **New Experiments for Model Scaling**
>
>   We added new experiments on **Qwen3-30B-A3B-Instruct (MOE)** and **Llama3.1-70B-Instruct** to explore scaling trend. MemArt matches or exceeds FullContext (up to **101.3\%**) and consistently outperforms baselines.
>
> * **Comparison with KV Sparse Method**
>
>   We clarified the fundamental difference between MemArt and KV sparse method and conducted  **H2O** as a representative KV sparse baseline.  MemArt improves accuracy by up to **23.9\%** under the same memory budget.
>
> * **Broader Dataset Scope**
>
>   We added evaluations on an additional long-horizon, multi-turn dataset, **PersonaMem** in *Appendix A.7*. Across models, MemArt consistently tracks FullContext and improves over Zep, Mem0, and H2O.
>
> * **Interpretability of retrieved KV blocks**
>
>   We added KV-to-text metadata and layer-wise relevance trajectories in *Appendix A.4*. Qualitative examples clearly show that the retrieved KV blocks align precisely with the correct and most relevant dialogue spans.
>
> * **Transferability to Closed-Source LLMs**
>
>   We clarified that stronger models typically yield more precise attention patterns, which directly enhance our attention-guided retrieval. The consistent improvement in performance observed across model scales (7B to 70B) validates that the mechanism benefits from the more precise attention patterns of larger models, supporting its application to closed-source LLMs.
>
> #### **Current Status**
>
> Reviewer 7mah initially gave high scores on both **soundness(4)** and **contribution(4)** , and their concerns primarily focused on the generality of our approach: larger model scales, new baselines, and broader datasets. Their follow-up question also centered on scaling to even larger models. Due to ICLR’s new policy, the reviewer was unable to provide feedback on our second-round response, but we strongly believe the extensive new experiments added during the rebuttal fully address all of their concerns and would likely lead to a more positive final assessment.
>
> ---
>
> ### Summary
>
> Through **extensive Empirical Validation**, including rigorous scaling up to 70B, the introduction of comprehensive baselines (H2O), robustness testing across diverse datasets (PersonaMem), enhanced interpretability, and quantified latency analysis; as well as **detailed Conceptual & Systemic Clarification**, featuring a clear technical distinction from sparse KV methods, clarification of storage assumptions, and MemArt’s role in multi-agent settings. We are confident that all reviewers’ core concerns have been thoroughly addressed. We thank all reviewers for their constructive input, which has meaningfully improved the rigor and clarity of the paper.
>
>
>
> Best regards,
>
> The authors of paper 11052

---

### Meta-Review · Area_Chair_eSP4 · 2025-12-19

**Summary:**

The paper proposes MemArt, a memory paradigm for LLM agents that transitions from plaintext storage to a Key-Value (KV) cache-centric approach. The authors introduce mechanisms such as AABB-based key compression, multi-token aggregation for retrieval, and decoupled positional encoding to enable the reuse of non-contiguous memory blocks. The primary goal is to mitigate the accuracy instability and prefix-caching disruptions associated with traditional plaintext memory systems.

**Reviewer Concerns:**

During the review process, the paper received ratings of 4, 6, and 4, indicating a generally borderline to negative reception. While reviewers acknowledged the novelty of operating in the latent space, significant concerns were raised regarding the practicality and scalability of the proposed system. Specifically, Reviewer ja5f highlighted the substantial storage overhead of storing float-based KV caches compared to plaintext, a fundamental bottleneck for long-horizon agents that the algorithmic contributions do not fully mitigate. Reviewer 7mah expressed concerns about the limited scope of the initial evaluation, particularly the reliance on smaller models and a single dataset, as well as the interpretability of retrieved latent blocks. Reviewer A26o pointed out potential system-level latencies involved in memory I/O and re-embedding.

After thorough evaluation, and having carefully read the original manuscript and each reviewer's comments and responses at least three times, I have decided to recommend rejection. Although the authors provided a comprehensive rebuttal (adding comparisons to H2O, results for larger models up to 70B parameters, and the PersonaMem dataset), the core reservations regarding the system's practicality remain. The massive storage footprint required for KV-centric memory presents a barrier to adoption that outweighs the demonstrated accuracy gains for many applications. Furthermore, the consensus among reviewers did not shift significantly following the rebuttal; the majority retained their initial scores, suggesting that the additional experiments, while valuable, did not sufficiently resolve the fundamental doubts about the method's efficiency and generalizability compared to optimized plaintext or RAG baselines. The paper presents an interesting concept, but the trade-offs involved require further justification and optimization to meet the bar for acceptance.

**Reviewer Scores:**

Reviewer ja5f (4): This reviewer would likely maintain their score, as their primary concern regarding the prohibitive storage growth of KV caches compared to plaintext is a fundamental architectural characteristic that was defended but not resolved.

Reviewer A26o (6): This reviewer would likely maintain their score, as they expressed satisfaction with the answers regarding latency but did not exhibit enough enthusiasm to champion the paper for a higher rating.

Reviewer 7mah (4): This reviewer would likely maintain their score; while the authors provided late-breaking results for 70B models, the initial lack of breadth and the reliance on open-source architectures left lingering doubts about the method's broad applicability.

---

### Decision · Program_Chairs · 2026-01-26

Reject